# Research on Very Volatile Organic Compounds and Odors from Veneered Medium Density Fiberboard Coated with Water-Based Lacquers

**DOI:** 10.3390/molecules27113626

**Published:** 2022-06-05

**Authors:** Weidong Wang, Xiwei Shen, Siqi Zhang, Ruixue Lv, Ming Liu, Wang Xu, Yu Chen, Huiyu Wang

**Affiliations:** 1Key Laboratory of Bio-Based Material Science and Technology, Ministry of Education, Northeast Forestry University, 26 Hexing Road, Harbin 150040, China; weidong_wang1024@163.com (W.W.); 987137216@nefu.edu.cn (S.Z.); 2016224287@nefu.edu.cn (R.L.); liuming1997@nefu.edu.cn (M.L.); xuwang@nefu.edu.cn (W.X.); chenyu@nefu.edu.cn (Y.C.); evelynlee@nefu.edu.cn (H.W.); 2School of Architecture, University of Nevada-Las Vegas, 4505 S. Maryland Pkwy, Las Vegas, NV 89154, USA

**Keywords:** very volatile organic compounds (VVOCs), odor, veneered medium fiber board (MDF), water-based lacquer, GC-MS/O

## Abstract

Very volatile organic compounds (VVOCs) are a group of important odor pollutants affecting indoor air quality that have been shown to be harmful to human health. A 15 L environmental chamber, combined with multi-bed tube was used to collect gases. Fifteen very volatile organic compounds (VVOCs), including 12 odor compounds, were identified from veneered medium density fiberboard coated with water-based lacquer (WB-MDF) using gas chromatography–mass spectrometry/olfactometry (GC-MS/O). The total very volatile organic compound (TVVOC) and total odor intensity (TOI) showed a decreasing trend over time, reaching equilibrium on day 28. TVVOC showed an overall slow-fast-slow emission profile, from day 3 to day 7, with a maximum decay rate of 29.7%. TOI showed the greatest rate of decline from day 1 to day 3, at approximately 12%. Alkane and alcohol VVOCs were the more abundant compounds, accounting for at least 60% and even up to 80% of the total. The major odor impression was fruity, with a highest odor rating of 6.6, followed by sweet, with an odor rating of 6.1. Although the odor impression changed from sweet to fruity over time, it seemed pleasant overall. The odor contributors were mainly alkanes, alcohols, esters, and ethers, which had relatively high odor intensities. The main odor-contributing substances were dichloromethane, ethanol, ethyl acetate, 2-methylacrylic acid methyl ester, and tetrahydrofuran. When WB-MDF is used for furniture or other decorative materials, it is strongly recommended that it be stored under ventilation for at least 28 days and the adoption of substitute solvents of lacquers, modified adhesives, and low-odor wood raw materials is recommended. These possible initiatives would contribute to the aim of building an environmentally friendly indoor environment.

## 1. Introduction

It is well known that indoor air quality (IAQ) is closely related to human comfort and health status. A non-polluted indoor environment helps to enhance the efficiency of people’s study and work, making our activities more economical and less time-consuming. Conversely, a polluted indoor space can slow people’s productivity and efficiency and even impair human health [1,2,3]. Additionally, odors can lead to multiple complaints from some occupants and seriously affect the quality of life.

Wooden furniture, which plays an increasingly important role in an interior environment, consists of solid wood, wood-based panels, lacquer-covered wood-based panels, and other decorative materials that can release various VOCs during long-term usage [4]. VOC emissions from wooden furniture are not the cumulative summation of the constituent materials, but are influenced not only by the structure of the material, but also by environmental conditions and interactions with other surrounding materials. Thus, VOC emission is an extremely complex process. The identification of the concentrations, composition, sources, and behaviors of VOCs is of great importance for developing effective abatement strategies. So far, a number of VOC works from wood-based panels have been investigated, involving improvements in sampling equipment and standard test methods that are essential to both academics and manufacturers [5], chemical components and emission levels [6], effects of environmental factors [7], the development of emission models to elucidate mechanisms [8,9,10], health risk assessment [11], and abatement control strategies [12].

Little effort has been dedicated to VVOCs in wood-based panels. VVOCs with a retention range below C_6_ have been proposed by the German Committee for Health-Related Evaluation of Building Product [13]. Because of their low boiling point, high volatility, and potentially strong carcinogenicity to humans, VVOCs should attract more attention. Schieweck A et al. performed small-volume sampling using tubes packed with Carbograph 5TD, in combination with thermal desorption-gas chromatography/mass spectrometry [14]. VVOCs between C_3_ and C_6_ could be detected even at very low limits of quantification. However, there were limitations for low-molecular-weight aldehydes and ketones (≤C_3_). Alexandra Schieweck found that C_4_ and C_5_ alkanes were VVOCs in wooden houses, which may originate from propellants in insulation materials. Furthermore, it was found that proper material selection remained important for achieving an acceptable indoor air quality [15]. The VVOC characteristics from *Choerospondias axillaris (Roxb) Burtt et Hill* coated with different lacquers were reported, with esters and alcohols determined to be the main emission substances. Ethyl acetate was determined to be the predominant odorous substance, coming from the solvent of ultraviolet paint [6].

Gas chromatography–mass spectrometry equipped with olfactometry detection (GC-MS/O) not only combines the excellent separation capability of gas chromatography with the rich structural information contained in mass spectrometry but also includes the sensitivity recognition of human olfactory detection, which has the potential to exceed the sensitivity of many chemical detectors. In recent years, with its unique advantages, this technique has been successfully used to select and evaluate comparatively important odor components from a complex mixture [16], such as tobacco, food aromas [17], flavorings and spices [18], medicines [19], and the environment [20]. In addition to these, this technology has been used in the wood and furniture industry. Wang et al. found more than 10 odor compounds from veneered particleboard coated with waterborne paint, identifying aromatics and alcohols as the major odor substances [21]. The odor profile of *Calocedrus decurrens (Torr.) Florin* was described via aroma extraction dilution analysis and two-dimensional GC-MS/O. In addition, 22 odor-active substances were identified, and five of these odorants were first reported as wood odorants, with thymoquinone proving to have a pencil-like odor [22]. Liu et al. found some unpleasant odors from aldehydes in wood-based panels, with octanal as the main odor source of these aldehydes [23]. Dong et al. investigated the odor characteristics of polyvinyl-chloride-overlaid medium density fiberboard (MDF). The major odor impressions were determined to be aromatic, acidic, and fresh, originating from toluene, ethylbenzene, phenanthrene, and dibutyl phthalate [24].

Lacquer-covered MDF is one of the most widely used materials for furniture and interior decoration in residential houses. This study was conducted to recognize the VVOCs, to identify the odor type, and to determine their intensity. At the same time, the possible emission sources of VVOC were traced back. This not only can supplement the database of volatile pollutants and provide a new understanding of low-molecular-substances from lacquered wood-based panels but also can help in selecting lacquer-covered materials for improved indoor air quality.

## 2. Results and Discussion

### 2.1. TVVOC and TOI of WB-MDF

Figure 1 shows the trends in total very volatile organic compound (TVVOC) and total odor intensity (TOI) of veneered MDF coated with water-based lacquer (WB-MDF). VVOC emissions of WB-MDF were a continuous process, and a time-dependent decrease in TVVOC was observed. These detected compounds were essentially emitted from the surface of the water-based lacquer in the early stages, with very small emissions from the substrate itself. During the initial emission phase (day 1), the TVVOC concentration was the highest, at 473.69 μg·m^−3^. With time, a decrease in TVVOC concentration was observed and this value was dropped to 442.69 μg·m^−3^, 310.83 μg·m^−3^, 268.62 μg·m^−3^, 223.91 μg·m^−3^ and 205.27 μg·m^−3^ on days 3, 7, 14, 21, and 28, decreasing by 7.01%, 34.38%, 43.29%, 52.73%, and 56.67%, respectively. From day 21 to day 28, the trend of TVVOC concentration decreased slowly and gradually reached an equilibrium TVVOC value. After 28 days of exposure, the TVVOC concentration from the WB-MDF reached an equilibrium level. In the initial emission phase, the TVVOC concentration value was approximately twice as high as the equilibrium TVVOC. The TVVOC concentration showed a smaller decay rate of only 7.01% during the first three days. Nevertheless, from day 3 to day 7, it displayed a maximum decay rate of 29.7% and showed a rapid evaporation process. Thereafter, the decay rate began to decline slowly at each phase, with all below 20%. The TVVOC decay rate was only 8.32% from day 21 to day 28. By the end of day 28, the VVOC emissions of WB-MDF gradually stabilized. The damping decrement showed an overall slow-rapid-slow phenomenon. At the same time, the TOI was found to show an emission profile similar to that of the TVVOC. Day 1 had the highest TOI of 15.2, which then dropped to 13.4, 13.1, 11.7, 10.8, and 10 on days 3, 7, 14, 21, and 28, decreasing by 11.80%, 13.81%, 23.02%, 28.95%, and 34.21%, respectively. From day 1 to day 3, the TOI revealed the greatest rate of decline, around 12%, while in other phases, it was lower. Taking into account the balance between the TVVOC, TOI, and the pollutant limiting value, it is highly recommended to increase the ventilation rate and the storage time to more than 28 days, which is conducive to improving the environmental properties of the panel. In addition, previous studies have found that adding some nanoparticles to coatings can reduce the emission of volatile pollutants, which may be effective for improving indoor air quality. Nanoparticles dispersed in the coatings can be embedded in the porous surface of the panels and act as photocatalysts and adsorbents for volatile pollutants [25]. The methods mentioned above are beneficial to improve the environmentally friendly properties of lacquered panels and to ensure human health.

### 2.2. VVOC Composition of WB-MDF

Figure 2 and Table 1 show the detailed VVOC components and their percentage content of WB-MDF. Our previous results showed that the multi-bed tubes presented excellent adsorption performance for alcohols and low-molecular-weight compounds and were able to capture some key compounds that could not be collected by the Tenax-TA tubes [26]. As shown in Figure 2 and Table 1, 15 VVOCs were detected from WB-MDF and the major VVOC components were divided into eight groups, consisting of alkanes (three substances), alcohols (three substances), esters (two substances), ketones (one substance), aldehydes (one substance), ethers (one substance), acids (two substances), and others (two substances). Among those compounds, alcohol and alkane VVOCs had relatively high concentration values, followed by ether and ester VVOCs. The concentrations of ketone and other VVOCs were relatively low and not significantly different.

For the initial emission stage (Figure 2a), alkanes were the most abundant emission components, accounting for more than 75% of the TVVOC concentration, followed by alcohols and ethers, which constituted more than 15% of the total concentration. Esters and ketones accounted for less than 3% of the total TVVOC concentration. Alcohols, esters, and ethers were partially derived from MDF and the veneered surface, and another large part came from the solvents, cosolvents, and auxiliary additives used in the production of the water-based lacquers.

The formation of alcohols and esters is complicated and probably results from comparatively complex chemical reactions. Furthermore, these two types of substances are partially derived from water-based lacquers and other additives. For example, ethanol is commonly used as a solvent for adhesives and 1-butanol could result from hydrolysis reactions of urea-formaldehyde resins [27]. In addition, 1,2-propanediol is commonly added to lacquers as a film-forming additive. Ethyl acetate is partially derived from the complex chemical reactions in cellulose and hemicellulose during hot pressing and from continuous emissions of ester compounds during the curing of the lacquer film. Meanwhile, 2-methylacrylic acid methyl ester has been traditionally used as a synthetic raw material for lacquers.

This study found that the dichloromethane contained in alkanes reached 324.51 μg·m^−3^, which made up more than 65% of the total concentration. This substance acts as an organic solvent; it is typically used for dilution and then evaporates. Its use requires care and attention, because this substance is on the list of toxic and hazardous air pollutants and has been classified as a group II (A) carcinogen. A high concentration of dichloromethane is irritating to the skin and mucous membranes, potentially damaging to the central nervous and respiratory systems, and even carcinogenic. By the end of day 3 (Figure 2b), alkanes showed a clear decreasing trend. However, alcohols, esters, and substances in the ‘others’ category showed the opposite trend and began to increase multiplicatively. Taking esters and ethers as examples, their concentrations ranged from 9.84 μg·m^−3^ and 27.23 μg·m^−3^ to 41.85 μg·m^−3^ and 117.06 μg·m^−3^, respectively, which represented a more than fourfold increase. Moreover, no new substances were released. From day 7 to day 14 (Figure 2c,d), alcohols were the most abundant substances, accounting for more than 60% of the total and even up to 80%. This resulted from the combination of the amounts of paneling and lacquer. Another possible reason for this abundance was that most solvents and the cosolvents of water-based lacquers were rapidly released in this phase. Ketones, esters, and ethers were mostly derived from solvents and auxiliaries, and some auxiliary additives were contained in both MDF adhesives and water-based lacquers but can only harm people at very high concentrations. In addition, the concentration of alkanes (including dichloromethane and 2,3-epoxy-2-methylbutane) dropped to zero and they were significantly less harmful. However, on day 14, a new substance, acetaldehyde, was detected with a concentration of 17.74 μg·m^−3^. Acetaldehyde is known to be a highly volatile, group II carcinogen whose risk to human health cannot be ignored and must be closely monitored. Some low molecular weight aldehydes and ketones may come from solvent residues, additives, and un-reacted raw materials, mainly by diffusion through evaporation [25]. Low concentrations of acetaldehyde could cause irritation of the eyes, nose, and upper respiratory tract, as well as bronchitis. High concentrations of acetaldehyde may cause headaches, drowsiness, confusion, bronchitis, pulmonary oedema, diarrhea, proteinuria, fatty degeneration of the liver and heart muscle, and even death in extremely severe cases.

As shown in Figure 2e,f, the main substances released from WB-MDF did not change significantly during the later emission stages; alcohols were still the predominant substances, and they constituted 38.3% of the totals on day 21 and up to 70% on day 28. Previous studies have reported that the main emission components of water-based lacquers are alcohols, esters, and aromatic hydrocarbon compounds [28]. Similarly, on day 21, some new substances, butane and N, N-dimethylformamide, were found at concentrations of 85.65 and 4.86 μg·m^−3^, respectively. N, N-dimethylformamide solvents are used to dissolve the emulsion particles in lacquer film dispersion, making it easier to form the film. In addition, this substance promotes the flatness of the paint film and improves its quality. From day 21 to day 28, VVOC emissions gradually stabilized and the variations were insignificant. Alcohols, alkanes, esters, and ethers were the major substances. Because of their harmful characteristics and highly toxicity, appropriately increasing ventilation rates and adding placement time are highly recommended to ensure these substances evaporate as quickly as possible.

### 2.3. Identification of Odor Compounds of WB-MDF

In terms of odor, any odorous substance with an intensity greater than 1 was registered. The odor characteristics are shown in Table 2. This study successfully identified 12 odorants, consisting of alkanes (three substances), alcohols (two substances), esters (two substances), aldehydes (one substance), ethers (one substance), acids (one substance), and others (two substances). Most odor compounds were relatively weak and were basically of moderate intensity. Only three odorous substances had an intensity higher than or equal to 3, namely, dichloromethane (3.4), tetrahydrofuran (3.2), and ethanol (3.0). Certainly, as seen in Table 1 and Table 2, the concentration of ethanol was high, but its odor intensity was relatively weak. Conversely, the tetrahydrofuran had a lower concentration but a higher odor intensity. The intensity of a substance is largely affected by its concentration, and there is no exact correlation between the intensity and the concentration of odor compounds.

Based on the assessors’ identification, the odor characteristics of WB-MDF were described in Table 2. Dichloromethane (no. 1) was identified as having a sweet and chloroform-like odor, and a similar odor impression was reported by Rossberg M. (2011) [29]. Research has shown that 2,3-epoxy-2-methylbutane (no. 2) has a sweet odor. Butane (no. 10) had an unpleasant and irritating odor, similar to the identification result of Lewis [30]. Ethanol (no. 3) was perceived as having an alcohol-like odor, which was coincided with the CAMEO chemical hazardous materials database. Meanwhile, 1-butanol (no. 7) was perceived as having a mildly alcohol-like odor, which was consistent with an earlier observation of the U.S. National Institute for Occupational Safety and Health (NIOSH, 1997). In addition, this substance was reported as having a sweet odor by Verschueren K. (2001) [31]. Ethyl acetate (no. 4) was reported to have a fruity odor, and a similar conclusion was reported by Fahlbusch K.G. (2003). 2-methylacrylic acid methyl ester (no. 5), was perceived to be a pungent odor. Tetrahydrofuran (no. 6) was reported to have a fruity odor. Acetaldehyde (no. 9) was perceived as having a fruity odor, whereas it was also reported to have a pungent odor. A sour odor for acetic acid (no. 11) was reported and 1,4-dioxane (no. 8) was considered to have a sweet odor. N, N-dimethylformamide (no. 12) was shown to have a fish-like odor, whereas it also presented a faint and amine-like odor, as reported by NIOSH (2010). As seen from the odor characteristics of the preceding compounds, we discovered that a single substance may have different odors and that the discrepancies among odors were considerable and perhaps even antagonistic. The odor characters are strongly related not only to the concentration but also to the material medium. These problems should also be taken into account.

The formation of odor compounds was very complex, and different odor compounds may interact with each other. There were four ways to affect the interaction between odor components in a mixture, namely, integration, synergism, antagonism, and independence [32]. To consider the complexities of the simultaneous existence of multiple odors, the integration effect was used in this odor analysis based on earlier findings [22,26]. The odor characteristics of WB-MDF were divided into six groups, namely, sweet, alcohol-like, fruity, irritating, sour, and fishy.

Figure 3 shows the odor radar profile of WB-MDF. As shown in Figure 3a, four odor characteristics appeared on day 1. Sweet was the dominant odor impression, with a high odor level of 6.1, followed by fruity at 4.5, and both contributed decisively and predominantly to the formation of the overall odor. Moreover, alcohol-like and irritating shared the same odor profile; both showed a fundamental complementary contribution with an odor rating of 2.3. On day 3, the major odor characteristic was altered and fruity was the major odor impression, with an odor rating of 5.9, playing a crucial contributory role to the formation of the overall odor profile. It was followed by alcohol-like (2.8). The other two odors, sweet and pungent, with an intensity of about 2, contributed a complementary modulating function. However, from day 3 to day 7, the alcohol-like odor became dominant, with a continuous decrease in intensity.

It can be seen from Figure 3b that fruity was the predominant odor during the post-emission stages. Furthermore, the contribution of alcohol-like and irritating odors was of great importance, with an odor rating around 2.0. However, on day 21, there was a noticeable difference with a significantly lower odor profile. A multi-odor mixture appeared in WB-MDF. At the same time, two unpleasant odors, sour and fishy, appeared in WB-MDF. This was mainly attributed to the appearance of two substances, acetic acid and N, N-dimethylformamide. Even if the intensity of their odors was not strong, they merited attention and concern.

Figure 3c shows the variability of the odor profile of WB-MDF. The odor profile showed a shrinking tendency at different testing periods. On the first day, the odor impressions were mainly sweet and fruity, and the main odor-contributing substances were dichloromethane, 2,3-epoxy-2-methylbutane, ethyl acetate, and tetrahydrofuran. On the 14th day, the odor impressions were mainly fruity and alcohol-like, and their main odor-contributing substances were ethyl acetate, tetrahydrofuran, acetaldehyde, and ethanol. On the 28th day, the main odor impression was fruity, and the intensity of alcohol-like and irritating was around 2. The main odor-contributing substances were ethyl acetate, tetrahydrofuran and acetaldehyde. The odor impressions from WB-MDF were rated as comforting, meaning that most people felt comfortable when in this impression environment. It also showed that the WB-MDF can be recommended as the material for interior furniture decoration and other applications.

Alkanes, esters, ethers, and alcohols were the main odor contributors with higher intensity values. Some of these compounds were derived from MDF, wood veneer, and adhesives, but most came from water-based lacquers and other additives contained in paints. When WB-MDF is used for furniture and other decorative applications, it is strongly recommended that it undergo at least 28 days of exposure to ensure that the contaminants dissipate. High ventilation rates are one option for speeding dissipation of the pollutants. When both energy consumption and air quality requirements are fully taken into account, an optimum air exchange rate might be confirmed, which is economically viable. More importantly, to obtain more environmentally friendly panels, partial substitution of tetrahydrofuran and N, N-dimethylformamide, synthetics, modified adhesives, and low-odor wood raw materials would be applicable. These possible initiatives would contribute to the goal of a healthy indoor environment.

## 3. Materials and Methods

### 3.1. Materials

The 18 mm thick original MDF was produced by a well-known furniture manufacturer in Guangzhou, China, and its formaldehyde (HCHO) emission level was E1 grade (≤1.5 mg/L). More detailed information about MDF is provided below. The original dimensions were 1200 mm × 1200 mm × 18 mm, and eucalyptus was used as the main raw material for its production. The density and moisture content ranged from 0.7 g/cm^3^ to 0.8 g/cm^3^ and 8% to 12%, respectively. The hot-pressing temperature was between 180 °C and 230 °C. The glue used for MDF production was urea-formaldehyde resin. The secondary processed specimen dimensions were 400 mm × 400 mm × 18 mm, after which 0.25 mm thick *Fraxinus mandshurica Rupr* veneers were glued to the both surfaces of the specimen on a hot press using urea-formaldehyde resin and polyvinyl acetate adhesive with a mass ratio of 6:4. The amount of glue used for one side of the veneer was 150 g/m^2^. The hot-pressing temperature, time, and pressure of the specimens were 100 °C, 3 min, and 1 MPa, respectively. The sample, measuring 150 mm × 75 mm × 18 mm, was prepared in a wood factory. A self-adhesive aluminum foil purchased from the local market was affixed to the edges of the samples to avoid gas volatility. Subsequently, the samples were coated with water-based lacquers. The parameters for the water-based lacquer are listed as follows: Xinletian, Shanghai Lixia Decoration Materials Co., transparent undercoat/twilight gray topcoat, main paint: diluent (ultra-pure water) = 10:1, painted two layers of undercoat (150 g/m^2^/session) and two layers of topcoat (150 g/m^2^/session), at least 12 h between painting sessions. The lacquer-covered sample, namely WB-MDF, was used as the testing material. The WB-MDF was stored in an environmental chamber or freely placed in the same environment as the experimental conditions, avoiding cross-contamination with other samples. Moreover, the WB-MDF was relocated in the environmental chamber at least 3 days prior to the next sampling.

### 3.2. Sampling

According to GB/T 29899-2013, a 15 L environmental chamber with controlled temperature and humidity conditions was used for gas sampling and proved to correlate well with a 1-m^3^ climate chamber [33]. This chamber, consisting mainly of glass material and silicone tubes, did not release any volatile components that could interfere with the testing. The schematic representation of a 15 L environmental chamber is shown in Figure 4. The relative humidity (RH) of the chamber air was adjusted by allowing a portion of the airflow to bubble through distilled water in a glass bottle at a controlled temperature. An automatic digital temperature and humidity sensor continuously monitored the temperature and humidity of the chamber. The temperature and humidity sensors provided a measurement accuracy and precision of 0.1 °C and 0.1%, respectively. Ultra-purity nitrogen gas (purity ≥ 99.999%), purchased from Harbin Liming Gas Company as the carrier gas, was continuously injected into the chamber for the purpose of exchange with the external environment. The measuring temperature of the sample in the chamber was determined to be 23 ± 1 °C and the relative humidity to be 50% ± 5%.

Before the chamber was loaded with the testing sample, the chamber’s inner surfaces were cleaned with hot water with neutral unscented detergent added, then rubbed with hot water and finally scrubbed at least two times using distilled water to remove any residual impurities, then dried, closed, and purged with nitrogen for at least 1 h. It was necessary to perform blank sampling to determine the minimum background concentration. These blank sampling tubes had sufficiently low background concentrations and not contain the target compound. Otherwise, the sampling chamber must be re-cleaned again until the quantity is considered to be acceptable. Each measured sample was then rapidly placed on a central iron holder in the chamber with a total exposed area of 0.0225 m^2^ and a loading rate (the ratio of the total sample exposed area to the environmental chamber volume) of 1.5 m^2^/m^3^. Ultra-purity nitrogen was continuously injected into the chamber through a glass rotor flowmeter at a constant airflow rate of 0.25 L/min. The metal electric fan mounted on the top of the chamber was then turned on and maintained until the testing was completed. Thus, the air in the chamber could be blended well. The sample was circulated in the chamber for 3.5 h before sampling.

The multi-bed tube, packed with carbopack C, carbopack B, and carboxen 1000, was used for gas sampling. Prior to sampling, these tubes were pre-treated at four temperature points (100 °C, 200 °C, 300 °C, and 380 °C) using a small thermal desorption processor (TP-2040; Beijing Beifen Tianpu Instrument Technology Co., Beijing, China) with ultra-purity nitrogen flow rates ranging from 0.05 L/min to 0.10 L/min, but usually 0.07 L/min. The pre-treated time was set to 15 min at one temperature point. The purpose of pre-treatment was to remove water vapor and residual impurities or gaseous contaminants from the tubes. To avoid interference from external factors and to ensure consistent sampling times, we collected samples at 3 pm on days 1, 3, 7, 14, 21, and 28. The sampling flow rate was set at 0.25 L/min, and the sampling time was 12 min. Thus, in total 3 L gas was collected in each tube by using a small vacuum pump (ANJ6513, Chengdu Xinweicheng Technology Co., Chengdu, Sichuan, China). When the gas sampling was finished, the two ends of the tube must be promptly sealed with matching copper caps supplied by the adsorption tube manufacturer. The collected gas samples were in principle immediately available for subsequent analysis. If the collected gas samples could not be used for analysis in time, they needed to be frozen in an empty refrigerator at −30 °C to prevent evaporation before further analysis, but no longer than 6 h.

### 3.3. Analytical Method for GC-MS

If necessary, the tubes were thawed at a normal temperature for more than 30 min prior to analysis. The external standard method was used directly in this experiment. The gas mixture was analyzed by thermal desorption-gas chromatography-mass spectrometry (TD-GC-MS) and quantified according to the relevant Chinese National Standard GB/T 29899-2013. The standard samples of benzene, toluene, ethylbenzene, styrene, naphthalene, and other target monomer compounds were prepared at concentrations of 10 μg/mL, 50 μg/mL, 200 μg/mL, 500 μg/mL, and 1000 μg/mL with the solvent of methanol. Based on the mass and peak area of the standard sample in the sorbent tube, a linear calibration equation, as shown in Equation (1), was obtained by the least square method. (The linear correlation coefficient γ^2^ should be greater than 0.995).
Mi = (Ai − bi)/Ki(1)
where Ai is the chromatographic peak area of the target compound_i_ in the adsorption tube of the standard sample, Ki is the gradient of the linear calibration equation of the target compound_i_, Mi is the mass of target compound_i_ in the adsorption tube (μg), and bi is the intercept of the linear calibration equation for the target compound_i_ on the Y-axis, which should be as small as possible.

The concentration of the target compound was calculated using Equation (2).
Ci = (Mi − Mi’)/V(2)
where Ci is the concentration of the target compound_i_ in the adsorption tube (μg/m^3^), Mi is the mass of the target compound_i_ in the adsorption tube (μg), Mi’ is the mass of the target compound_i_ in the blank adsorption tube after sampling (μg), V is the sampling volume (m^3^).

The concentrations of the other compounds were calculated from the linear calibration equation for toluene.

The thermal desorption unit was produced by Markes International Ltd. (Llantrisant, UK) and it was operated at an analysis temperature of 300 °C and a pipeline temperature of 180 °C. The dry purge, thermal desorption, and the trap desorption were held for 5 min, 10 min, and 5 min, respectively. Ultra-pure helium (purity ≥ 99.999%) was used as the carrier gas for the GC at a constant flow velocity of 1 mL/min.

The DSQ II series quadrupole gas chromatography and mass spectrometry (GC-MS) was produced by Thermo Fisher Scientific Ltd. (Waltham, MA, USA), and a nonpolar or slightly polar DB-5MS GC column supplied by Agilent Technologies had a specification of 30 m × 0.25 mm × 0.25 μm and had a stationary phase of 5% phenyl and 95% polysiloxane. The GC column oven was programmed initially at 40 °C and held for 2 min, then ramped up to 50 °C at 2 °C/min and held at that temperature for 4 min, then ramped up to 150 °C at 5 °C/min and held at 150 °C for 2 min, then ramped up to 250 °C at 10 °C/min, and finally held at 250 °C for 8 min. The composition of the desorbed gases was accurately identified based on their retention times (RT) and a mass spectrometry detector (MS), and compared with the National Institute of Standards and Technology (NIST) and Wiley libraries (only matched substances with a positive or negative degree between 750 and 1000). The MS detector had a full scan mode with an electron impact (MS-EI) energy of 70 eV. The mass-to-charge ratio ranged from 40 to 450 amu. The transmission line and ion source temperatures were maintained at 270 °C and 230 °C, respectively. The Xcalibur database was performed to record real-time data and subsequently analyzed. The relative percentage content of each VVOC of WB-MDF component was obtained by the area normalization method.

Repeated experiments were conducted in triplicate and the mean VVOC concentration was then calculated from the duplicated measurements. The total very volatile organic compound (TVVOC) was derived by summing the concentrations of odorous and non-odorous substances. The total odor intensity (TOI) was the summation of the intensities of all odorous substances.

### 3.4. Analytical Method for GC-O

The sniffer 9100 olfactory port produced by Brechbuhler (Schlieren, Switzerland) was used in this study. The transfer line was heated to avoid condensation of the analytes on the capillary walls. A split valve was attached to the end of the GC column and two capillary columns were attached to the split valve. The gas mixture was separated by GC after thermal desorption and part of it went to the mass spectrometer for identification and the rest to the sniffer for sniffing operation. The GC effluents from the GC column were split 1:1. The auxiliary gas (humid air) supplied by a water bottle was continuously added to the sniffer port to reduce damage to the nasal mucosa caused by dryness during prolonged analysis for the odor assessors. The schematic representation of gas chromatography-mass spectrometry coupled with the olfactory detector is shown in Figure 5. This test procedure is referenced elsewhere in the literature [34]. The detailed screening and training recommendations were in accordance with ISO 12219-7-2017: five trained and experienced odor assessors, aged between 20 and 30, without relevant olfactory disorders and non-smokers, were selected to form an odor assessment group to identify and evaluate the odor. All assessors were already screened for sensitivity, motivation, ability to concentrate, and ability to remember and identify odor characteristics. After a series of professional training sessions, the assessors were well versed in various odor characteristics and the assessment method. At the same time, to receive accurate odor data from the testing samples, some actions such as eating or drinking strong stimulating foods and chewing gum were strictly prohibited for 5 h prior to taking part in this GC-O experiment. In addition, they were not allowed to wear heavy cosmetics and perfumes, or strong deodorants on the day of the olfactory assessment. When the GC operation system started running, the odor assessors recorded the real-time data based on their odor characteristics and intensity values as well as the retention times. The total run time for the GC system was 53 min. A maximum time of 50 min was recommended for sniffing, as continuous sniffing may lead to human nose fatigue. The direct intensity method was chosen in this sniffing experiment according to the Japanese Standards (Ministry of the Environment, 1971); the odor intensity was judged on a scale from 0 to 5, with 0 = none, 1 = very weak, 2 = weak, 3 = moderate, 4 = strong, and 5 = very strong. When the same odor characteristics were described by at least two assessors, the odor identification results were formally adopted. The odor intensity value was obtained from the average values of different assessors. The identification of the odor substances was based on the GC-O method and compared with other references.

According to EN 13725-2003 (NSAI 2003), the sniffer laboratory required a temperature of 21 °C–23 °C, a relative humidity of 40%, good ventilation, and no other special odors during the entire sniffer operation.

## 4. Conclusions

In this study, a 15 L environmental chamber, combined with muti-bed tubes, was used for gas sampling. Very volatile organic compounds (VVOCs) and odors from veneered medium density fiberboard coated with water-based lacquer (WB-MDF) were investigated using gas chromatography-mass spectrometry/olfactometry (GC-MS/O). In total 15 VVOCs, including 12 odor compounds, were successfully identified and were divided into alkanes, alcohols, esters, ketones, aldehydes, ethers, acids, and other substances. The total very volatile organic compounds (TVVOCs) and total odor intensity (TOI) of WB-MDF showed a decreasing trend over time, reaching equilibrium on day 28. TVVOC showed an overall slow-fast-slow emission profile, from day 3 to day 7, with a maximum decay rate of 29.7%. TOI showed the greatest rate of decline from day 1 to day 3, at approximately 12%. Alkanes and alcohols were the more abundant compounds, accounting for at least 60% and even up to 80% of the total concentration. The dominant odor impressions of WB-MDF were sweet and fruity, which played a decisive factor in the formation of the overall odor. The major odor contributors were found to be alkanes, esters, ethers and alcohols. The main odor-contributing substances were dichloromethane, ethanol, ethyl acetate, 2-methylacrylic acid methyl ester, and tetrahydrofuran, and their main odor impressions were sweet, alcohol-like, fruity, and irritant. On day 21 and day 28, butane and N, N-dimethylformamide, were to be found, and both contributed to unpleasant odors that should be taken seriously. When WB-MDF is used for furniture and other decorative applications, a minimum exposure of 28 days is strongly recommended to ensure dissipation of contaminants. It is also important to develop awareness about increasing ventilation when using lacquer-covered wood-based panels, as this can decrease volatile organic pollutant levels in indoor air. High ventilation rates are one option for accelerating the airing-out of the pollutants. Even more importantly, to obtain more environmental panels, partial substitutions of tetrahydrofuran and N, N-dimethylformamide, synthetics, modified adhesives, and low-odor wood raw materials are applicable. These possible initiatives would contribute to the goal of an environmentally friendly indoor environment.

## Figures and Tables

**Figure 1 molecules-27-03626-f001:**
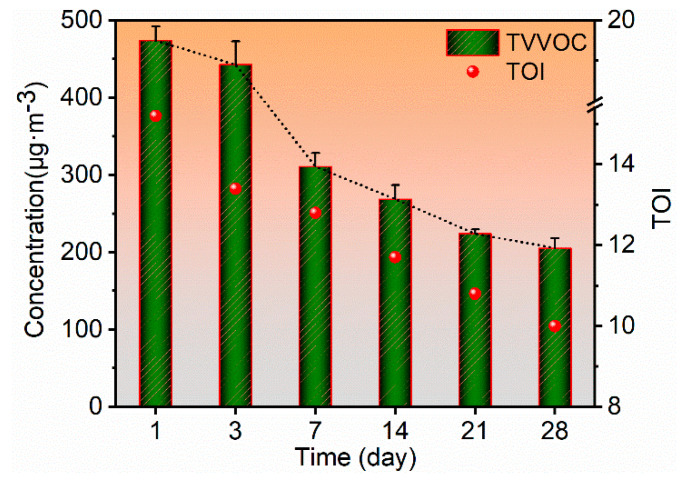
TVVOC and TOI of WB-MDF.

**Figure 2 molecules-27-03626-f002:**
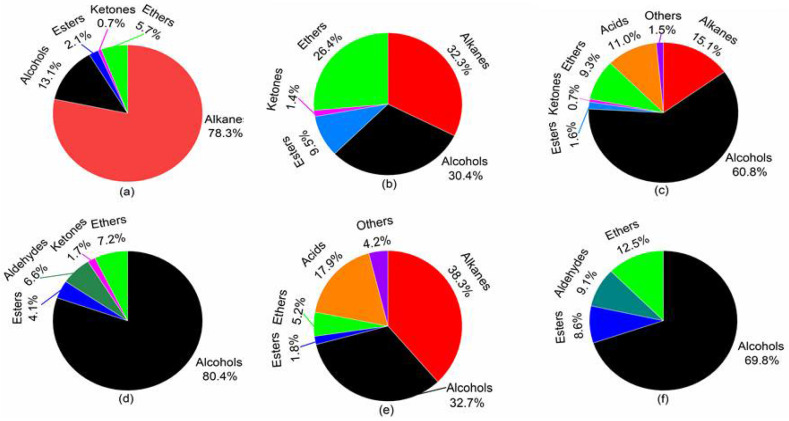
The VVOC compositions and their percentage content of WB-MDF. (**a**) day 1; (**b**) day 3; (**c**) day 7; (**d**) day 14; (**e**) day 21; (**f**) day 28.

**Figure 3 molecules-27-03626-f003:**
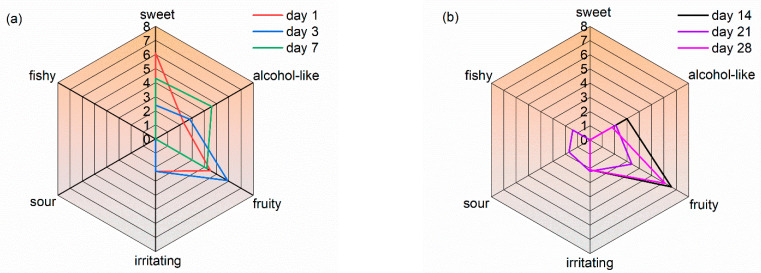
The odor profile spectra of WB-MDF at different testing times. (**a**) the odor profile spectra on days 1, 3, 7; (**b**) the odor profile spectra on days 14, 21, 28; (**c**) the variability of odor profile spectra on days 1, 14, 28.

**Figure 4 molecules-27-03626-f004:**
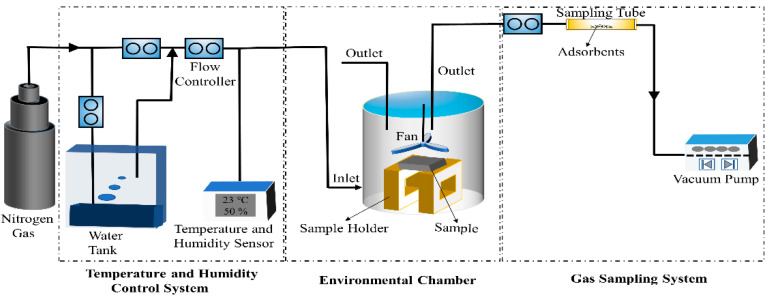
The schematic representation of the 15 L environmental chamber.

**Figure 5 molecules-27-03626-f005:**
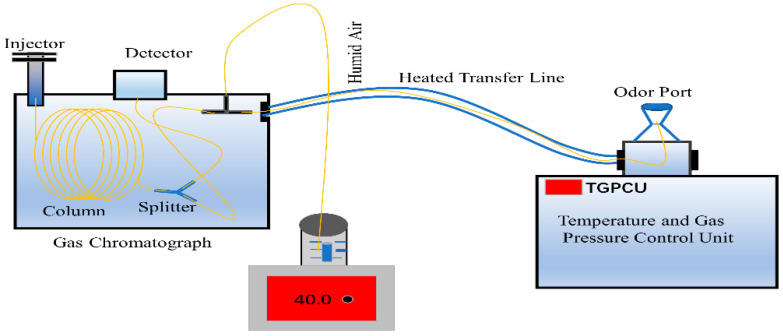
The schematic representation of gas chromatography-mass spectrometry coupled with olfactory detector.

**Table 1 molecules-27-03626-t001:** The VVOC components of WB-MDF in detail.

Group	Number	Boiling Point (°C)	Compound	Mass Concentration (μg·m^−3^)					
				1	3	7	14	21	28
**Alkanes**	1	75.0	2,3-Epoxy-2-methylbutane	46.61	/	46.94	/	/	/
	2	39.8	Dichloromethane	324.51	143.17	/	/	/	/
	3	−0.5	Butane	/	/	/	/	85.65	/
	Total			371.12	143.17	46.94	/	85.65	/
**Alcohols**	4	78.3	Ethanol	62.01	134.46	86.19	170.29	67.51	57.97
	5	117.6	1-Butanol	/	/	102.64	/	/	/
	6	184.8	1,2-Propanediol	/	/	/	45.59	5.81	85.86
	Total			62.01	134.46	188.83	215.88	73.32	143.83
**Esters**	7	100.0	2-Methylacrylic acid methyl ester	5.03	10.93	/	4.78	/	8.16
	8	77.2	Ethyl acetate	4.81	30.92	4.97	6.36	3.96	9.49
	Total			9.84	41.85	4.97	11.14	3.96	17.65
**Aldehydes**	9	20.8	Acetaldehyde	/	/	/	17.74	/	18.7
	Total			/	/	/	17.74	/	18.7
**Ketones**	10	222.4	3-Methyl-2(5H)-furanone	3.49	6.15	2.15	4.52	/	/
	Total			3.49	6.15	2.15	4.52	/	/
**Ethers**	11	66.0	Tetrahydrofuran	27.23	117.06	29.02	19.34	11.59	25.79
	Total			27.23	117.06	29.02	19.34	11.59	25.79
**Acids**	12	149.0	N-Formylglycine	/	/	34.14	/	26.06	/
	13	117.9	Acetic acid	/	/	/	/	13.91	/
	Total			/		34.14	/	39.97	/
**Others**	14	153.0	N, N-Dimethylformamide	/	/	/	/	4.86	/
	15	101.0	1,4-Dioxane	/	/	4.78	/	4.56	/
	Total			/	/	4.78	/	9.32	/
**TVVOC**				473.69	442.69	310.83	268.62	223.91	205.27

/, not detected.

**Table 2 molecules-27-03626-t002:** The odor characteristics and intensity levels of WB-MDF.

No.	Chemical Formula	Compound	RI	Odor Characteristic	Odor Intensity					
					1	3	7	14	21	28
**1**	CH_2_Cl_2_	Dichloromethane	<600	sweet	3.4	2.4	x	x	x	x
**2**	C_5_H_10_O	2,3-Epoxy-2-methylbutane	<600	sweet	2.7	x	2.5	x	x	x
**3**	C_2_H_6_O	Ethanol	<600	alcohol-like	2.3	2.8	2.5	3.0	2.1	1.8
**4**	C_4_H_8_O_2_	Ethyl acetate	<600	fruity	1.8	2.7	1.8	2.1	1.6	1.8
**5**	C_5_H_8_O_2_	2-Methylacrylic acid methyl ester	701	pungent, irritant	2.3	2.3	x	2.1	x	2.1
**6**	C_4_H_8_O	Tetrahydrofuran	627	fruity, ether-like	2.7	3.2	2.4	2.3	1.8	2.1
**7**	C_4_H_10_O	1-Butanol	647	alcohol-like	x	x	2.1	x	x	x
**8**	C_4_H_8_O_2_	1,4-Dioxane	700	sweet	x	x	1.8	x	x	x
**9**	C_2_H_4_O	Acetaldehyde	<600	fruity	x	x	x	6	x	2.2
**10**	C_4_H_10_	Butane	<600	irritating	x	x	x	x	2.2	x
**11**	C_2_H_4_O_2_	Acetic acid	<600	sour, vinegar-like	x	x	x	x	1.7	x
**12**	C_3_H_7_NO	N, N-Dimethylformamide	772	fishy	x	x	x	x	1.4	x

RI, retention index; x, not identified.

## Data Availability

All data generated or analyzed during this study are included in this article.

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
