# Peer review of "Research on Very Volatile Organic Compounds and Odors from Veneered Medium Density Fiberboard Coated with Water-Based Lacquers"

_molecules, 2022, doi:10.3390/molecules27113626_

Round 1

Reviewer 1 Report

Line 225:  "...it was also presented a faint and amine-like odor.....". The word "was" must be eliminated.

Line 273: "...at least days of exposu28 re to ensure...." Please correct the typing. 

Author Response

Dear editors and reviews,

Thank you very much for taking time to review the previous version of our manuscript. The revised manuscript, with more accurate, standardized, complete and detailed expression of content, follows the principle of truth and factuality in scientific and academic research. In addition, your valuable suggestions enable us to improve our work.

We submit the revised version in the annex, and use ' Track Changes ' to facilitate your view. At the same time, we are also very grateful to Ms. Julie Laing for her assistance with language.

Please see the attachment for the specific amendments and responses to your suggestions.

Thank you again for your help.

Best regards,

Reviewer 2 Report

Wang et al. performed study on very volatile organic compounds and odors from veneered medium density fiberboard.  Authors have been carried analysis using a 15L chamber and sampled VVOC using cartridges including adsorbents at day 1, day 3, day 7, day 21 and day 28.

Authors found the presence of 15 VVOC including 12 odor compounds presented in table 1. In this table lack the ebullition temperature in order to clarify the classification of authors of dioxane as VVOC. In addition, in the manuscript authors did not precise the acquisition time per analysis and there is a lack in the method development especially if authors carried out a blank analysis between tuns.

A major problem of this study, is in the discussion part. No background analysis was carried out in order to understand the efficiency of the sampling method, authors must address comparison of studies carried in literature.

Author Response

(The authors gave the same response as above.)

Reviewer 3 Report

In the manuscript (molecules-1716134), entitled “Research on Very Volatile Organic Compounds and Odors from Veneered Medium Density Fiberboard coated with Water-based Lacquers” The emission of fifteen very volatile organic compounds (VVOCs) from veneered medium density fiberboard coated with water-based lacquer (WB-MDF) were monitored using gas chromatography–mass spectrometry/olfactometry (GC-MS/O). The changes in total odor intensities, the major odor impressions, and also in each analyte's concentrations were monitored over time. This manuscript is generally well written and it can be reconsidered after major revision. Some points are listed below.

1)            Please revise the following sentence

Line 308: The metal electric fan mounted on the top of the chamber was then turned on and held until the end of the testing.

2)            Line 60: Reference 14 should be cited at the end of the following sentence.

“Schieweck A et al. performed small-volume sampling using tubes packed with Carbograph 5TD, in combination with thermal desorption–gas chromatography/mass spectrometry.”

3)            “Time (d)” should be corrected as “Time (day)” in the x-axis of Figure 1

4)            The method for the calculation of the concentration of analytes should be defined in more detail in the manuscript.

5)            Please don't use the "d" abbreviation instead of the day.

6)            Please don’t use Fig. abbreviation (Line 136: Fig. 2a, Line 159: Fig 2.b) instead of Figure.

7)            Qualification of analytes directly from the MS library can cause errors. Have the authors validated the detected compounds with their standards?

8)            Please change the location of the title below from 195 to 196 "3.3. Identification of odor compounds of WB-MDF" .

9)            The list of identified VVOC and odor chemicals can be presented as a table.

10)          The storage conditions of the samples before and during the sampling steps should be presented in the manuscript.

11)          Line 304: Please place the following number in a single line  if it is possible

Line 304: “2.25 × 10-2 m2”.

12)          There are some points to clarify in the sampling step. Was the nitrogen passage from the environmental chamber continuing in the adsorption step of analytes to the sorption tube (steady-state or dynamic conditions)? How long the samples have stayed in the environmental chamber (3h or 5h?), and when the sampling was carried out in the sorption tube?

13)          Figure 4. should be revised. All the nitrogen gas passed through the circulation system seems to pass over the adsorption tube. In addition, please indicate the cell in Figure 4 as an environmental chamber instead of a circulation system.

14)          Line 338: Please use mL/min unit for “0.001 L/min”.

15)          Line 341: Please use “30 m” instead of “30,000 mm”.

16)          What is the length of two capillary columns in the following sentence.

Line 367: "A split valve was attached to the end of the GC column and two capillary columns were attached to the split valve."

17)          The humid air connection part to the olfactometer should be checked in Figure 5.

18)          Signal generator in Figure 5 should be explained in the manuscript.

19)          Line 421: Please use "day" instead of "d" abbreviation.

20)         Author should discuss the results with the following papers.

ZHU, X. D.; LIU, Y.; SHEN, J. Volatile organic compounds (VOCs) emissions of wood-based panels coated with nanoparticles modified water based varnish. European Journal of Wood and Wood Products, 2016, 74.4: 601-607.

ULKER, Ozge Cemiloglu; ULKER, Onur; HIZIROGLU, Salim. Volatile organic compounds (VOCs) emitted from coated furniture units. Coatings, 2021, 11.7: 806.

Author Response

(The authors gave the same response as above.)

Round 2

Reviewer 3 Report

In the current version of the manuscript (molecules-1716134) the required corrections have been made and it can be acceptable in its current form. Please control the following point.  

 Line 164: Please check the compond's name. (methy 2-methyl-2-propenoate)